METHODS AND RESOURCES

# Deep learning allows genome-scale prediction of Michaelis constants from structural features

**Alexander Kroll**[1], **Martin K. M. Engqvist**[2], **David Heckmann**[1]*, **Martin J. Lercher**[1]*

**1** Institute for Computer Science and Department of Biology, Heinrich Heine University, Düsseldorf, Germany, **2** Department of Biology and Biological Engineering, Chalmers University of Technology, Gothenburg, Sweden

* david.heckmann@hhu.de (DH); martin.lercher@hhu.de (MJL)

**Data Availability Statement:** All datasets generated and the Python code used to produce the results (in Jupyter notebooks) are available from https://github.com/AlexanderKroll/KM_prediction.

## Abstract

The Michaelis constant $K_M$ describes the affinity of an enzyme for a specific substrate and is a central parameter in studies of enzyme kinetics and cellular physiology. As measurements of $K_M$ are often difficult and time-consuming, experimental estimates exist for only a minority of enzyme–substrate combinations even in model organisms. Here, we build and train an organism-independent model that successfully predicts $K_M$ values for natural enzyme–substrate combinations using machine and deep learning methods. Predictions are based on a task-specific molecular fingerprint of the substrate, generated using a graph neural network, and on a deep numerical representation of the enzyme's amino acid sequence. We provide genome-scale $K_M$ predictions for 47 model organisms, which can be used to approximately relate metabolite concentrations to cellular physiology and to aid in the parameterization of kinetic models of cellular metabolism.

## Introduction

The Michaelis constant, $K_M$, is defined as the concentration of a substrate at which an enzyme operates at half of its maximal catalytic rate; it hence describes the affinity of an enzyme for a specific substrate. Knowledge of $K_M$ values is crucial for a quantitative understanding of enzymatic and regulatory interactions between enzymes and metabolites: It relates the intracellular concentration of a metabolite to the rate of its consumption, linking the metabolome to cellular physiology.

As experimental measurements of $K_M$ and $k_{cat}$ are difficult and time-consuming, no experimental estimates exist for many enzymes even in model organisms. For example, in *Escherichia coli*, the biochemically best characterized organism, in vitro $K_M$ measurements exist for less than 30% of natural substrates (see Methods, "Download and processing of $K_M$ values"), and turnover numbers have been measured in vitro for only about 10% of the approximately 2,000 enzymatic reactions [1].

$K_M$ values, together with enzyme turnover numbers, $k_{cat}$, are required for models of cellular metabolism that account for the concentrations of metabolites. The current standard approach in large-scale kinetic modeling is to estimate kinetic parameters in an optimization process

**Funding:** This work was funded through grants to M.J.L. by the Volkswagenstiftung (in the "Life?" program) and by the Deutsche Forschungsgemeinschaft (CRC 1310 and, under Germany's Excellence Strategy, EXC 2048/1, Project ID: 390686111). The funders had no role in study design, data collection and analysis, decision to publish, or preparation of the manuscript.

**Competing interests:** The authors have declared that no competing interests exist.

**Abbreviations:** D-MPNN, directed message passing neural network; DTBA, drug–target binding affinity; EC, Enzyme Commission; ECFP, extended connectivity fingerprint; FCNN, fully connected neural network; GNN, graph neural network; LogP, octanol–water partition coefficient; MRPE, mean relative prediction error; MSE, mean squared error; MW, molecular weight; QSAR, quantitative structure–activity relationship; ReLU, rectified linear unit.

[2–4]. These optimizations typically attempt to estimate many more unknown parameters than they have measurements as inputs, and, hence, the resulting $K_M$ and $k_{cat}$ values have wide confidence ranges and show little connection to experimentally observed values [2]. Therefore, predictions of these values from artificial intelligence, even if only up to an order of magnitude, would represent a major step toward more realistic models of cellular metabolism and could drastically increase the biological understanding provided by such models.

Only few previous studies attempted to predict kinetic parameters of natural enzymatic reactions in silico. Heckmann and colleagues [5] successfully employed machine learning models to predict unknown turnover numbers for reactions in *E. coli*. They found that the most important predictors of $k_{cat}$ were the reaction flux catalyzed by the enzyme, estimated computationally through parsimonious flux balance analysis, and structural features of the catalytic site. While many *E. coli* $k_{cat}$ values could be predicted successfully with this model, active site information was not available for a sizeable fraction of enzymes [5]. Moreover, neither active site information nor reaction flux estimates are broadly available beyond a small number of model organisms, preventing the generalization of this approach.

Borger and colleagues [6] trained a linear model to predict $K_M$ values based on other $K_M$ measurements for the same substrate paired with different enzymes in the same organism and with the same enzymes in other organisms; they fitted an independent model for each of 8 different substrates. Yan and colleagues [7] later followed a similarly focused strategy, predicting $K_M$ values of beta-glucosidases for the substrate cellobiose based on a neural network. These 2 previous prediction approaches for $K_M$ targeted individual, well-studied enzyme–substrate combinations with ample experimental $K_M$ data for training and testing. Their strategies are thus unsuitable for less well-studied reactions and cannot be applied to genome-scale predictions.

A related problem to the prediction of $K_M$ is the prediction of drug–target interactions, an important task in drug development. Multiple approaches for the prediction of drug–target binding affinities (DTBAs) have been developed (reviewed in [8]). Most of these approaches are either similarity-based, structure-based, or feature-based. Similarity-based methods rely on the assumption that similar drugs tend to interact with similar targets; these methods use known drug–target interactions to learn a prediction function based on drug–drug and target–target similarity measures [9,10]. Structure-based models for DTBA prediction utilize information on the target protein's 3D structure [11,12]. Neither of these 2 strategies can easily be generalized to genome-scale, organism-independent predictions, as many enzymes and substrates share only distant similarities with well-characterized molecules, and 3D structures are only available for a minority of enzymes.

In contrast to these first 2 approaches, feature-based models for drug–target interaction predictions use numerical representations of the drug and the target as the input of fully connected neural networks (FCNNs) [13–16]. The drug feature vectors are most often either SMILES representations [17], expert-crafted fingerprints [18–20], or fingerprints created with graph neural networks (GNNs) [21,22], while those of the targets are usually sequence-based representations. As this information can easily be generated for most enzymes and substrates, we here use a similar approach to develop a model for $K_M$ prediction.

An important distinction between the prediction of $K_M$ and DTBA prediction is that the former aims to predict affinities for known, natural enzyme–metabolite combinations. These affinities evolved under natural selection for the enzymes' functions, an evolutionary process strongly constrained by the metabolite structure. In contrast, wild-type proteins did not evolve in the presence of a drug, and, hence, molecular structures are likely to contain only very limited information about the binding affinity for a target without information about the target protein.

Despite the central role of the metabolite molecular structure for the evolved binding affinity of its consuming enzymes, important information on the affinity must also be contained in the enzyme structure and sequence. To predict $K_M$, it would be desirable to employ detailed structural and physicochemical information on the enzyme's substrate binding site, as done by Heckmann and colleagues for their $k_{cat}$ predictions in *E. coli* [5]. However, these sites have only been characterized for a minority of enzymes [23]. An alternative approach is to employ a multidimensional numerical representation of the entire amino acid sequence of the enzyme, as provided by UniRep [24]. UniRep vectors are based on a deep representation learning model and have been shown to retain structural, evolutionary, and biophysical information.

Here, we combine UniRep vectors of enzymes and diverse molecular fingerprints of their substrates to build a general, organism-, and reaction-independent model for the prediction of $K_M$ values, using machine and deep learning models. In the final model, we employ a 1,900-dimensional UniRep vector for the enzyme together with a task-specific molecular fingerprint of the substrate as the input of a gradient boosting model. Our model reaches a coefficient of determination of $R^2 = 0.53$ between predicted and measured values on a test set, i.e., the model explains 53% of the variability in $K_M$ values across different, previously unseen natural enzyme–substrate combinations. In **S1 Data**, we provide complete $K_M$ predictions for 47 genome-scale metabolic models, including those for *Homo sapiens*, *Mus musculus*, *Saccharomyces cerevisiae*, and *E. coli*.

## Results

For all wild-type enzymes in the BRENDA database [25], we extracted organism name, Enzyme Commission (EC) number, UniProt ID, and amino acid sequence, together with information on substrates and associated $K_M$ values. If multiple $K_M$ values existed for the same combination of substrate and enzyme amino acid sequence, we took the geometric mean. This resulted in a dataset with 11,675 complete entries, which was split into a training set (80%) and a test set only used for the final validation (20%). All $K_M$ values were $\log_{10}$-transformed.

### Predicting $K_M$ from molecular fingerprints

To train a prediction model for $K_M$, we first had to choose a numerical representation of the substrate molecules. For each substrate in our dataset, we calculated 3 different expert-crafted molecular fingerprints, i.e., bit vectors where each bit represents a fragment of the molecule. The expert-crafted fingerprints used are extended connectivity fingerprints (ECFPs), RDKit fingerprints, and MACCS keys. We calculated them with the python package RDKit [19] based on MDL Molfiles of the substrates (downloaded from KEGG [26]; a Molfile lists a molecule's atom types, atom coordinates, and bond types [27]).

MACCS keys are 166-dimensional binary fingerprints, where each bit contains the information if a certain chemical structure is present in a molecule, e.g., if the molecule contains a ring of size 4 or if there are fewer than 3 oxygen atoms present in the molecule [20]. RDKit fingerprints are generated by identifying all subgraphs in a molecule that do not exceed a particular predefined range. These subgraphs are converted into numerical values using hash functions, which are then used to indicate which bits in a 2,048-dimensional binary vector are set to 1 [19]. Finally, to calculate ECFPs, molecules are represented as graphs by interpreting the atoms as nodes and the chemical bonds as edges. Bond types and feature vectors with information about every atom are calculated (types, masses, valences, atomic numbers, atom charges, and number of attached hydrogen atoms) [18]. Afterwards, these identifiers are updated for a predefined number of steps by iteratively applying predefined functions to summarize aspects of neighboring atoms and bonds. After the iteration process, all identifiers are

used as the input of a hash function to produce a binary vector with structural information about the molecule. The number of iterations and the dimension of the fingerprint can be chosen freely. We set them to the default values of 3 and 1,024, respectively; lower or higher dimensions led to inferior predictions.

To compare the information on $K_M$ contained in the different molecular fingerprints independent of protein information, we used the molecular fingerprints as the sole input to elastic nets, FCNNs, and gradient boosting models. To the fingerprints, we added the 2 features molecular weight ($MW$) and octanol–water partition coefficient ($LogP$), which were shown to be correlated with the $K_M$ value [28]. The models were then trained to predict the $K_M$ values of enzyme–substrate combinations (**Fig** 1A). The FCNNs consisted of an input layer with the dimension of the fingerprint (including the additional features $MW$ and $LogP$), 2 hidden layers, and a 1D output layer (for more details, see Methods). Gradient boosting is a machine learning technique that creates an ensemble of many decision trees to make predictions. Elastic nets are regularized linear regression models, where the regularization coefficient is a linear combination of the $L_1-$ and $L_2$-norm of the model parameters. For each combination of the 3 model types and the 3 fingerprints, we performed a hyperparameter optimization with 5-fold

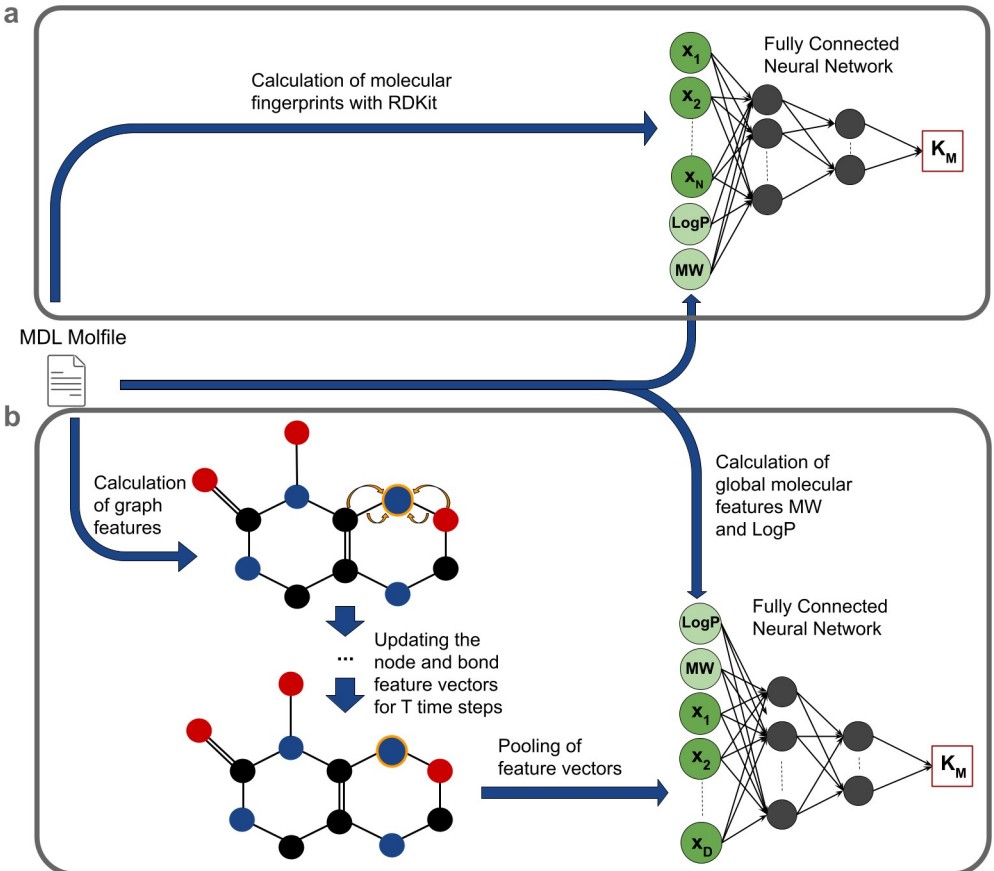

**Fig 1. Model overview. (a)** Predefined molecular fingerprints. Molecular fingerprints are calculated from MDL Molfiles of the substrates and then passed through machine learning models like the FCNN together with 2 global features of the substrate, the $MW$ and $LogP$. **(b)** GNN fingerprints. Node and edge feature vectors are calculated from MDL Molfiles and are then iteratively updated for $T$ time steps. Afterwards, the feature vectors are pooled together into a single vector that is passed through an FCNN together with the $MW$ and $LogP$. FCNN, fully connected neural network; GNN, graph neural network; $LogP$, octanol–water partition coefficient; $MW$, molecular weight.

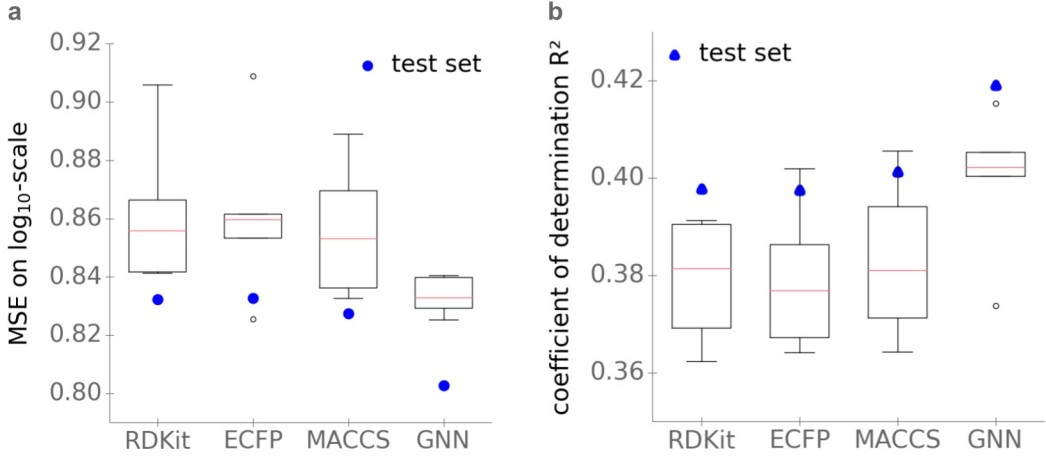

**Fig 2. When using only substrate features as inputs, task-specific molecular fingerprints (GNN) lead to better $K_M$ predictions than predefined, expert-crafted fingerprints.** (a) *MSE* on $\log_{10}$-scale. **(b)** Coefficients of determination $R^2$. Boxplots summarize the results of the 5-fold cross-validations on the training set; blue dots show the results on the test set. The data underlying the graphs shown in this figure can be found at https://github.com/AlexanderKroll/KM_prediction/tree/master/figures_data. ECFP, extended connectivity fingerprint; GNN, graph neural network; MSE, mean squared error.

cross-validation on the training set, measuring performance through the mean squared error (MSE). For all 3 types of fingerprints, the gradient boosting model outperformed the FCNN and the elastic net (**S1–S3 Tables**).

The $K_M$ predictions with the gradient boosting model based solely on the substrate ECFP, MACCS keys, and RDKit molecular fingerprints showed very similar performances on the test set, with $MSE = 0.83$ and coefficients of determination $R^2 = 0.40$ (**Fig 2**).

## Best $K_M$ predictions from metabolite fingerprints using graph neural networks and gradient boosting

Recent work has shown that superior prediction performance can be achieved through task-specific molecular fingerprints, where a deep neural network simultaneously optimizes the fingerprint and uses it to predict properties of the input. In contrast to conventional neural networks, these GNNs can process non-Euclidean inputs, such as molecular structures. This approach led to state-of-the-art performances on many biological and chemical datasets [21,22].

As an alternative to the predefined, expert-crafted molecular fingerprints, we thus also tested how well we can predict $K_M$ from a task-specific molecular fingerprint based on a GNN (**Fig 1**; for details, see Methods, "Architecture of the graph neural network"). As for the calculations of the ECFPs, each substrate molecule is represented as a graph by interpreting the atoms as nodes and the chemical bonds as edges, for which feature vectors are calculated from the MDL Molfiles. These are updated iteratively for a fixed number of steps, in each step applying functions with learnable parameters to summarize aspects of neighboring atoms and bonds. After the iterations, the feature vectors are pooled into 1 molecular fingerprint vector. In contrast to ECFPs, the parameters of the update functions are not fixed but are adjusted during the training of the FCNN that predicts $K_M$ from the pooled fingerprint vector (Methods). As for the predefined molecular fingerprints, we defined an extended GNN fingerprint by adding the 2 global molecular features *LogP* and *MW* to the model before the $K_M$ prediction step.

To compare the learned substrate representation with the 3 predefined fingerprints, we extracted the extended GNN fingerprint for every substrate in the dataset and fitted an elastic net, an FCNN, and a gradient boosting model to predict $K_M$. As before, we performed a hyperparameter optimization with 5-fold cross-validation on the training set for all models. The gradient boosting model again achieved better results than the FCNN and the elastic net (**S1–S3 Tables**). The performance of our task-specific fingerprints is better than that of the predefined fingerprints, reaching an $MSE = 0.80$ and a coefficient of determination $R^2 = 0.42$ on the test set, compared to an $MSE = 0.83$ and $R^2 = 0.40$ for the other fingerprints (**Fig 2**). To compare the performances statistically, we used a one-sided Wilcoxon signed-rank test for the absolute errors of the predictions for the test set, resulting in $p = 0.0080$ (ECFP), $p = 0.073$ (RDKit), and $p = 0.062$ (MACCS keys). While the differences in the error distributions are only marginally statistically significant for RDKit and MACCS keys at the 5% level, these analyses support the choice of the task-specific GNN molecular fingerprint for predicting $K_M$.

It is noteworthy that the errors on the test set are smaller than the errors achieved during cross-validation. We found that the number of training samples has a great influence on model performance (see below, "Model performance increases linearly with training set size"). Hence, the improved performance on the test set may result from the fact that before validation on the test set, models are trained with approximately 2,000 more samples than before each cross-validation.

## Effects of molecular weight and octanol–water partition coefficient

Before predicting $K_M$ from the molecular fingerprints, we added the $MW$ and the $LogP$. Do these extra features contribute to improved predictions by the task-specific GNN fingerprints? To answer this question, we trained GNNs without the additional features $LogP$ and $MW$, as well as with only one of those additional features. **Fig 3** displays the performance of gradient boosting models that are trained to predict $K_M$ with GNN fingerprints with and without extra features, showing that the additional features have only a small effect on performance: Adding both features reduces MSE from 0.82 to 0.80, while increasing $R^2$ from 0.41 to 0.42. The

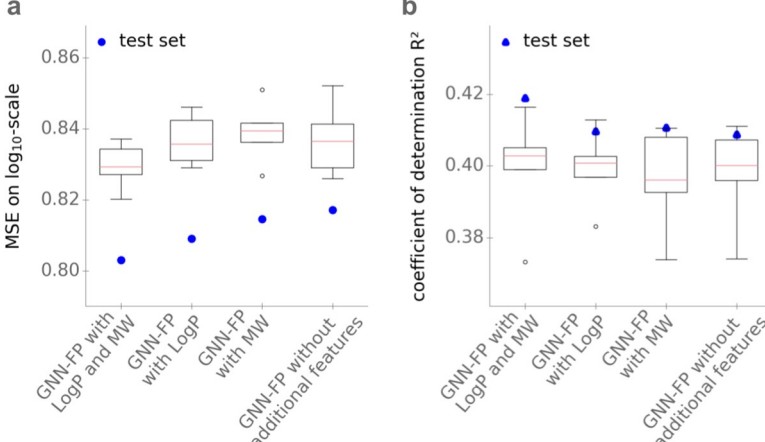

**Fig 3. Adding *MW* and *LogP* as features has only a minor effect on the performance of the GNN in predicting $K_M$.** (**a**) *MSE* on log₁₀-scale. (**b**) Coefficients of determination $R^2$. Models use the GNN with additional features *LogP* and *MW*; with only one of the additional features; and without the 2 features. Boxplots summarize the results of the 5-fold cross-validations on the training set; blue dots show the results on the test set. The data underlying the graphs shown in this figure can be found at https://github.com/AlexanderKroll/KM_prediction/tree/master/figures_data. GNN, graph neural network; *LogP*, octanol–water partition coefficient; MSE, mean squared error; *MW*, molecular weight.

difference in model performance is not statistically significant ($p = 0.13$, one-sided Wilcoxon signed-rank test for the absolute errors of the predictions for the test set). This indicates that most of the information used to predict $K_M$ can be extracted from the graph of the molecule itself. However, since the addition of the 2 additional features slightly improves $K_M$ predictions on the test dataset, we include the features *MW* and *LogP* in our further analyses.

## UniRep vectors as additional features

So far, we have only considered substrate-specific information. As $K_M$ values are features of specific enzyme–substrate interactions, we now need to add input features that represent enzyme properties. Important information on substrate binding affinity is contained in molecular features of the catalytic site; however, active site identities and structures are available only for a small minority of enzymes in our dataset.

We thus restrict the enzyme information utilized by the model to a deep numerical representation of the enzyme's amino acid sequence, calculating an UniRep vector [24] for each enzyme. UniRep vectors are 1,900-dimensional statistical representation of proteins, created with an mLSTM, a recurrent neural network architecture for sequence modeling that combines the long short-term memory and multiplicative recurrent neural network architectures. The model was trained with 24 million unlabeled amino acid sequences to predict the next amino acid in an amino acid sequence, given the previous amino acids [24]. In this way, the mLSTM learns to store important information about the previous amino acids in a numerical vector, which can later be extracted and used as a representation for the protein. It has been shown that these representations lead to good results when used as input features in prediction tasks concerning protein stability, function, and design [24].

## Predicting $K_M$ using substrate and enzyme information

To predict the $K_M$ value, we concatenated the 52-dimensional task-specific extended fingerprint learned with the GNN and the 1,900-dimensional UniRep vector with information about the enzyme's amino acid sequence into a global feature vector. This vector was then used as the input for a gradient boosting model for regression in order to predict the $K_M$ value. We also trained an FCNN and an elastic; however, predictions were substantially worse (**S4**–**S6 Tables**), consistent with the results obtained when using only the substrate fingerprints as inputs.

The gradient boosting model that combines substrate and enzyme information achieves an $MSE = 0.65$ on a $\log_{10}$-scale and results in a coefficient of determination $R^2 = 0.53$, substantially superior to the above models based on substrate information alone. We also validate our model with an additional metric, $r_m^2$, which is a commonly used performance measurement tool for quantitative structure–activity relationship (QSAR) prediction models. It is defined as $r_m^2 = r^2 \times (1 - \sqrt{r^2 - r_0^2})$, where $r^2$ and $r_0^2$ are the squared correlation coefficients with and without intercept, respectively [29,30]. Our model achieves a value of $r_m^2 = 0.53$ on the test set.

**Fig 4A** and **4B** compare the performance of the full model to models that use only substrate or only enzyme information as inputs, applied to the BRENDA test dataset (which only contains previously unseen enzyme–substrate combinations). To predict the $K_M$ value from only the enzyme UniRep vector, we again fitted a gradient boosting model, leading to $MSE = 1.01$ and $R^2 = 0.27$. To predict the $K_M$ value from substrate information only, we chose the gradient boosting model with extended task-specific fingerprints as its inputs, which was used for the comparison with the other molecular fingerprints. (**Fig 2**).

**Fig 4A** and **4B** also compare the 3 models to the naïve approach of simply using the mean over all $K_M$ values in the training set as a prediction for all $K_M$ values in the test set, resulting in

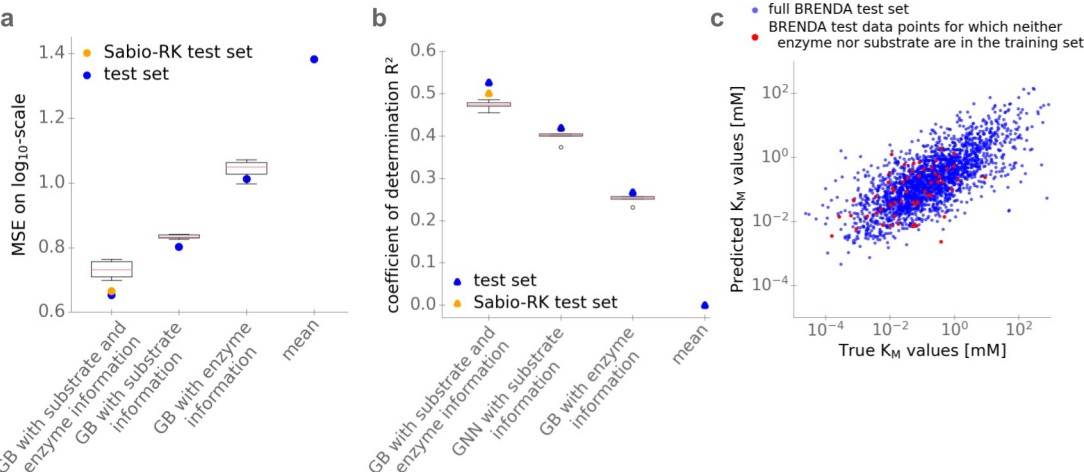

**Fig 4. Performance of the optimized models. (a)** *MSE*. **(b)** Coefficients of determination ($R^2$). Values in (a) and (b) are calculated using the gradient boosting model with different inputs: substrate and enzyme information; substrate information only (GNN); and enzyme information only (domain content). Boxplots summarize the results of the 5-fold cross-validations on the training set; blue dots show the results on the test set. For comparison, we also show results on the test set from a naïve model using the mean of the $K_M$ values in the training set for all predictions. **(c)** Scatter plot of $\log_{10}$-transformed $K_M$ values of the test set predicted with the gradient boosting model with substrate and enzyme information as inputs versus the experimental values downloaded from BRENDA. Red dots are for combinations where neither enzyme nor substrate were part of the training set. The data underlying the graphs shown in this figure can be found at https://github.com/AlexanderKroll/KM_prediction/tree/master/figures_data. GB, gradient boosting; GNN, graph neural network; MSE, mean squared error.

$MSE$ = 1.38 and $R^2$ = 0. **Fig 4C** compares the values predicted using the full model with the experimental values of the test set obtained from BRENDA.

## Predicting $K_M$ for an independently acquired test dataset

Our model was trained and tested on data from BRENDA. To confirm its prediction power, it is desirable to test it on data from other sources. We thus created an additional, independent test set by obtaining the same type of information from the Sabio-RK database [31], keeping only entries that were not already included in the BRENDA dataset. This resulted in a second test set with 274 entries. The model trained on the BRENDA data achieves a very similar performance ($MSE$ = 0.67, $R^2$ = 0.49) on the independent Sabio-RK test data (orange dots in **S1 Fig**).

## Predicting $K_M$ for enzymes and substrates not represented in the training data

Homologous enzymes that catalyze the same reaction tend to have broadly similar kinetic parameters. To test to what extent such similarities affect our results, we investigated how well our model performs for the 664 data points in the test set that have substrate–EC number combinations not found in the training set (violet dots in **S1 Fig**). The $K_M$ predictions for these data points resulted in an $MSE$ = 0.79 and $R^2$ = 0.45, compared to $MSE$ = 0.65 and $R^2$ = 0.53 for the full test data.

It is conceivable that predictions are substantially better if the training set contains entries with the same substrate or with the same enzyme, even if not in the same combination. In practice, one may however want to also make predictions for combinations where the enzyme and/or the substrate are not represented in the training data at all. To test how our model performs in such cases, we separately analyzed those 57 entries in the test data where neither

enzyme nor substrate occurred in the training data, resulting in $MSE = 0.74$ and $R^2 = 0.26$, compared to $MSE = 0.65$ and $R^2 = 0.53$ for the full test data (red points in **Fig 4C**). At least in part, the smaller $R^2$ value can be explained by the poor predictions for $K_M$ values below $10^{-2}$ $mM$ (see the residuals in panel **a** in **S2 Fig**). The training dataset contained few $K_M$ values in this region (panel **b** in **S2 Fig**)—there may have been too little training data here for the challenging task of predicting $K_M$ for unseen enzymes and substrates. In contrast, the model performs substantially better for unseen substrates and enzymes with $K_M$ values between $10^{-2}$ and $10^0$ $mM$, where much more training data were available. We conclude that given enough training data, the proposed model appears capable of predicting $K_M$ values also for data points where substrate and/or enzyme are not in the training set.

## Model performance increases linearly with training set size

The last analysis indicates that prediction performance may be strongly affected by the amount of relevant training data. Indeed, the training datasets employed for AI prediction tasks are typically vastly larger than those available for predicting $K_M$. To test if the size of the training set has a substantial, general effect on prediction quality, we trained the final gradient boosting model with different amounts of the available training samples. We excluded randomly data points from the original training set for this analysis, creating 6 different training sets with sizes ranging from about 4,500 to approximately 9,500 data points. **Fig 5** shows that model performance—measured either in terms of $MSE$ or $R^2$—increases approximately linearly with the size of the training set. This result indicates that our models are still far from overfitting and that increasing availability of data will allow more accurate predictions in the future.

## $K_M$ predictions for enzymatic reactions in genome-scale metabolic models

Above, we have described the development and evaluation of a pipeline for genome-scale, organism-independent prediction of $K_M$ values. This pipeline and its parameterization can be used, for example, to obtain preliminary $K_M$ estimates for enzyme–substrate combinations of interest or to parameterize kinetic models of enzymatic pathways or networks. To facilitate such applications, we predicted $K_M$ values for all enzymes in 47 curated genome-scale metabolic models, (**S1 Data**), include models for *E. coli*, *S. cerevisiae*, *M. musculus*, and *H. sapiens*.

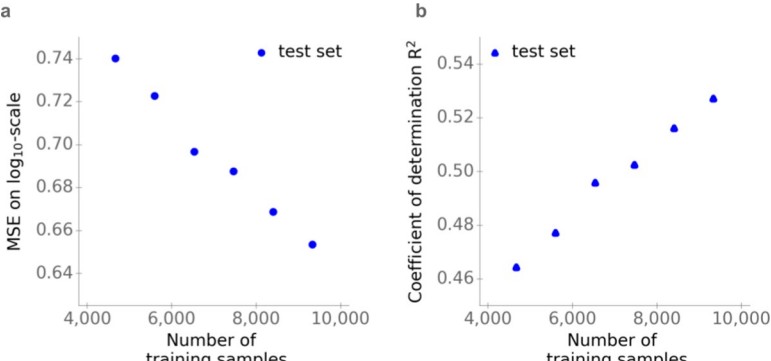

**Fig 5. Effect of the training set size on model performance. (a)** *MSE*. **(b)** Coefficients of determination ($R^2$). Values in (a) and (b) are calculated for the test sets, using the gradient boosting model with substrate and enzyme information as the input. The gradient boosting model is trained with different amounts of the available training samples. The data underlying the graphs shown in this figure can be found at https://github.com/AlexanderKroll/KM_prediction/tree/master/figures_data. MSE, mean squared error.

These models are for organisms from different domains, while the training and test data are dominated by bacteria. To test if this uneven training data distribution leads to biases, we divided our test set into subsets belonging to the domains Archaea, Bacteria, and Eukarya, calculating separate $MSE$ and $R^2$ values for each domain. The test set contained 142 data points from Archaea, with $MSE = 0.71$ and $R^2 = 0.37$; 1,439 data points from Bacteria, with $MSE = 0.65$ and $R^2 = 0.51$; and 749 data points from Eukarya, with $MSE = 0.64$ and $R^2 = 0.56$. We therefore conclude that our model can predict $K_M$ values for different domains approximately equally well.

The predictions for the genome-scale metabolic models in **S1 Data** are based on a machine learning model trained with all of the available data, including all data points from the test set. For 73% of the reactions across all 47 metabolic models, substrate and enzyme information were available, such that the full prediction model could be applied. For 15% only substrate information, for 10% only enzyme information, and for 2% neither substrate nor enzyme information were available. We treated situations with missing information as follows: If information on only one of the 2 molecules (enzyme or substrate) was available, we used the corresponding reduced prediction model (with either only UniRep vector or only extended GNN representation as input, respectively). If both substrate and enzyme information were missing, we predicted the $K_M$ value as the geometric mean of all $K_M$ values in our dataset.

## Discussion

In conclusion, we found that Michaelis constants of enzyme–substrate pairs, $K_M$, can be predicted through artificial intelligence with a coefficient of determination of $R^2 = 0.53$: More than half of the variance in $K_M$ values across enzymes and organisms can be predicted from deep numerical representations of enzyme amino acid sequence and substrate molecular structure. This performance is largely organism-independent and does not require that either enzyme or substrate are covered by the dataset used for training; the good performance was confirmed using a second, independent and nonoverlapping test set from Sabio-RK ($R^2 = 0.49$). To obtain this predictive performance, we used task-specific fingerprints of the substrate (GNN) optimized for the $K_M$ prediction, as these appear to contain more information about $K_M$ values than predefined molecular fingerprints based on expert-crafted transformations (ECFP, RDKit fingerprint, MACCS keys). The observed differences between GNNs and predefined fingerprints is in line with the results of a previous study on the prediction of chemical characteristics of small molecules [22].

**Fig 4**, which compares $K_M$ predictions across different input feature sets, indicates that the relevant information contained in an enzyme's amino acid sequence may be less important for its evolved binding affinity to a natural substrate than the substrate's molecular structure: Predictions based only on substrate structures explain almost twice as much variance in $K_M$ compared to predictions based only on enzyme representations. It is possible, though, that improved (possibly task-specific) enzyme representations will modify this picture in the future.

A direct comparison of the prediction quality of our model to the results of Yan and colleagues [7] would not be meaningful, as the scope of their model is very different from that of ours. Yan and colleagues trained a model specific to a single enzyme–substrate pair with only 36 data points, aiming to distinguish $K_M$ values between different sequences of the same enzyme (beta-glucosidase) for the same substrate (cellobiose). However, the performance of our general model, with $MSE = 0.65$, compares favorably to that of the substrate-specific statistical models of Borger and colleagues [6], which resulted in an overall $MSE = 1.02$.

We compare our model to 2 different models for DTBA prediction, DeepDTA and SimBoost [10,16]. These two, which were trained and tested on the same 2 datasets, achieved $r_m^2$

values ranging from 0.63 to 0.67 on test sets. This compares to $r_m^2 = 0.53$ achieved for $K_M$ predictions with our approach. It is generally difficult to compare prediction performance between models trained and tested on different datasets. Here, this difficulty is exacerbated by the different prediction targets (DTBA versus $K_M$). Crucially, the datasets used for DTBA and $K_M$ prediction differ substantially with respect to their densities, i.e., the fraction of possible protein–ligand combinations covered by the training and test data. One of the datasets used for DTBA prediction encompasses experimental data for all possible drug–target combinations between 442 different proteins and 68 targets (442×68 = 30,056). The second dataset contains data for approximately 25% of all possible combinations between 229 proteins and 2,111 targets (118,254 out of 229×2,111 = 483, 419). In contrast, our $K_M$ dataset features 7,001 different enzymes and 1,582 substrates but comprises only about 0.1% of their possible combinations (11,600 out of 7,001×1582 = 11,075, 582). Thus, our dataset is not only much smaller, but also has an extremely low coverage of possible protein–ligand combinations compared to the DTBA datasets used in [10,16]. As shown in **Fig 5**, the number of available training samples has a strong impact on model performance, and the same is likely true for the data density. Against this background, the performance of our $K_M$ prediction model could be seen as being surprisingly good. **Fig 5** indicates that $K_M$ predictions can be improved substantially once more training data become available.

To provide the model with information about the enzyme, we used statistical representations of the enzyme amino acid sequence. We showed that these features provide important enzyme-specific information for the prediction of $K_M$. It appears likely that predictions could be improved further by taking features of the enzyme active site into account—such as hydrophobicity, depth, or structural properties [5]—once such features become widely available [23]. Adding organism-specific information, such as the typical intracellular pH or temperature, may also increase model performance.

We wish to emphasize that our model is trained to predict $K_M$ values for enzyme–substrate pairs that are known to interact as part of the natural cellular physiology, meaning that their affinity has evolved under natural selection. The model should thus be used with care when making predictions for enzyme interactions with other substrates, such as nonnatural compounds or substrates involved in moonlighting activities. In such cases, DTBA prediction models (with their higher data density) may be better suited, and estimates with our model should be regarded as a lower bound for $K_M$ that might be reached under appropriate natural selection.

To put the performance of the current model into perspective, we consider the mean relative prediction error *MRPE* = 4.1, meaning that our predictions deviate from experimental estimates on average by 4.1-fold. This compares to a mean relative deviation of 3.4-fold between a single $K_M$ measurement and the geometric mean of all other measurements for the same enzyme–substrate combination in the BRENDA dataset (the geometric means of enzyme–substrate combinations were used for training the models). Part of the high variability across values in BRENDA is due to varying assay conditions in the in vitro experiments [28]. Moreover, entries in BRENDA are not free from errors; on the order of 10% of the values in the database do not correspond to values in the original papers, e.g., due to errors in unit conversion [28].

Especially on the background of this variation, the performance of our enzyme–substrate specific $K_M$ model appears remarkable. In contrast to previous approaches [6,7,13–16], the model requires no previous knowledge about measured $K_M$ values for the considered substrate or enzyme. Furthermore, only one general purpose model is trained, and it is not necessary to obtain training data and to fit new models for individual substrates, enzyme groups, or

organisms. Once the model has been fitted, it can provide genome-scale $K_M$ predictions from existing features within minutes. We here provide such predictions for a broad set of model organisms, including mouse and human; these data can provide base estimates for unknown kinetic constants, e.g., to relate metabolomics data to cellular physiology, and can help to parameterize kinetic models of metabolism. Future work may develop similar prediction frameworks for enzyme turnover numbers ($k_{cat}$), which would facilitate the completion of such parameterizations.

## Methods

### Software and code availability

We implemented all code in Python [32]. We implemented the neural networks using the deep learning library TensorFlow [33] and Keras [34]. We fitted the gradient boosting models using the library XGBoost [35].

All datasets generated and the Python code used to produce the results (in Jupyter notebooks) are available from https://github.com/AlexanderKroll/KM_prediction. Two of the Jupyter notebooks contain all the necessary steps to download the data from BRENDA and Sabio-RK and to preprocess it. Execution of a second notebook performs training and validation of our final model. Two additional notebooks contain code to train the models with molecular fingerprints as inputs and to investigate the effect of the 2 additional features, *MW* and *LogP*, for the GNN.

### Downloading and processing $K_M$ values from BRENDA

We downloaded $K_M$ values together with organism and substrate name, EC number, UniProt ID of the enzyme, and PubMed ID from the BRENDA database [25]. This resulted in a dataset with 156,387 entries. We mapped substrate names to KEGG Compound IDs via a synonym list from KEGG [26]. For all substrate names that could not be mapped to a KEGG Compound ID directly, we tried to map them first to PubChem Compound IDs via a synonym list from PubChem [36] and then mapped these IDs to KEGG Compound IDs using the web service of MBROLE [37]. We downloaded amino acid sequences for all data points via the UniProt mapping service [38] if the UniProt ID was available; otherwise, we downloaded the amino acid sequence from BRENDA via the organism name and EC number.

We then removed (i) all duplicates (i.e., entries with identical values for $K_M$, substrate, and amino acid sequence as another entry); (ii) all entries with non-wild-type enzymes (i.e., with a commentary field in BRENDA labeling it as mutant or recombinant); (iii) entries for nonbacterial organisms without an UniProt ID for the enzyme; and (iv) entries with substrate names that could not be mapped to a KEGG Compound ID. This resulted in a filtered set of 34,526 data points. Point (iii) was motivated by the expectation that isoenzymes are frequent in eukaryotes but rare in bacteria, such that organism name and EC number are sufficient to unambiguously identify an amino acid sequence in the vast majority of cases for bacteria but not for eukaryotes. If multiple $\log_{10}$-transformed $K_M$ values existed for 1 substrate and 1 amino acid sequence, we took the geometric mean across these values. For 11,737 of these, we could find an entry for the EC number–substrate combination in the KEGG reaction database. Since we are only interested in $K_M$ values for natural substrates, we only kept these data points [28]. We $\log_{10}$-transformed all $K_M$ values in this dataset. We split the final dataset with 11,737 entries randomly into training data (80%) and test data (20%). We further split the training set into 5 subsets, which we used for 5-fold cross-validations for the hyperparameter optimization of the machine learning models. We used the test data to evaluate the final models after hyperparameter optimization.

To estimate the proportion of metabolic enzymes with $K_M$ values measured in vitro for *E. coli*, we mapped the *E. coli* $K_M$ values downloaded from BRENDA to reactions of the genome scale metabolic model *i*ML1515 [39], which comprises over 2,700 different reactions. To do this, we extracted all enzyme–substrate combinations from the *i*ML1515 model for which the model annotations listed an EC number for the enzyme and a KEGG Compound ID for the substrate, resulting in 2,656 enzyme–substrate combinations. For 795 of these combinations (i.e., 29.93%), we were able to find a $K_M$ value in the BRENDA database.

## Download and processing of $K_M$ values from Sabio-RK

We downloaded $K_M$ values together with the name of the organism, substrate name, EC number, UniProt ID of the enzyme, and PubMed ID from the Sabio-RK database. This resulted in a dataset with 8,375 entries. We processed this dataset in the same way as described above for the BRENDA dataset. We additionally removed all entries with a PubMed ID that was already present in the BRENDA dataset. This resulted in a final dataset with 274 entries, which we used as an additional test set for the final model for $K_M$ prediction.

## Calculation of predefined molecular fingerprints

We first represented each substrate through 3 different molecular fingerprints (ECFP, RDKit fingerprint, MACCS keys). For every substrate in the final dataset, we downloaded an MDL Molfile with 2D projections of its atoms and bonds from KEGG [26] via the KEGG Compound ID. We then used the package Chem from RDKit [19] with the Molfile as the input to calculate the 2,048-dimensional binary RDKit fingerprints [19], the 166-dimensional binary MACCS keys [20], and the 1,024-dimensional binary ECFPs [18] with a radius of 3.

## Architecture of the fully connected neural network with molecular fingerprints

We used an FCNN to predict $K_M$ values using only representations of the substrates as input features. We performed a 5-fold cross-validation on the training set for each of the 4 substrate representations (ECFP, RDKit fingerprints, MACCS keys, and task-specific fingerprints) for the hyperparameter optimization. The FCNN consisted of 2 hidden layers, and we used rectified linear units (*ReLUs*), which are defined as $ReLU(x) = max(x, 0)$, as activation functions in the hidden layers to introduce nonlinearity. We applied batch normalization [40] after each hidden layer. Additionally, we used *L2*-regularization in every layer to prevent overfitting. Adding dropout [41] did not improve the model performance. We optimized the model by minimizing the *MSE* with the stochastic gradient descent with Nesterov momentum as an optimizer. The hyperparameters regularization factor, learning rate, learning rate decay, dimension of hidden layers, batch size, number of training epochs, and momentum were optimized by performing a grid search. We selected the set of hyperparameters with the lowest mean *MSE* during cross-validation. The results of the cross-validations and best set of hyperparameters for each fingerprint are displayed in **S1 Table**.

## Fitting of the gradient boosting models with molecular fingerprints

We used gradient boosting models to predict $K_M$ values using only representations of the substrates as input features. As for the FCNNs, we performed a 5-fold cross-validation on the training set for each of the 4 substrate representations (ECFP, RDKit fingerprints, MACCS keys, and task-specific fingerprints) for hyperparameter optimization. We fitted the models using the gradient boosting library XGBoost [35] for Python. The hyperparameters

regularization coefficients, learning rate, maximal tree depth, maximum delta step, number of training rounds, and minimum child weight were optimized by performing a grid search. We selected the set of hyperparameters with the lowest mean *MSE* during cross-validation. The results are displayed in **S2 Fig**.

### Fitting of the elastic nets with molecular fingerprints

We used elastic nets to predict $K_M$ values with representations of the substrates as input features. Elastic nets are linear regression model with additional *L1*- and *L2*-penalties for the coefficients of the model in order to apply regularization. We performed 5-fold cross-validations on the training set for all 4 substrate representations (ECFP, RDKit fingerprints, MACCS keys, and task-specific fingerprints) for hyperparameter optimization. During hyperparameter optimization, the coefficients for *L1*-regularization and *L2*-regularization were optimized by performing a grid search. The models were fitted using the machine learning library scikit-learn [42] for Python. The results of the hyperparameter optimizations are displayed in **S3 Table**.

### Calculation of molecular weight (*MW*) and the octanol–water partition coefficient (LogP)

We calculated the additional 2 molecular features, *MW* and *LogP*, with the package Chem from RDKit [19], with the MDL Molfile of the substrate as the input.

### Calculation of the input of the graph neural network

Graphs in GNNs are represented with tensors and matrices. To calculate the input matrices and tensors, we used the package Chem from RDKit [19] with MDL Molfiles of the substrates as inputs to calculate 8 features for very atom $v$ (atomic number, number of bonds, charge, number of hydrogen bonds, mass, aromaticity, hybridization type, chirality) and 4 features for every bond between 2 atoms $v$ and $w$ (bond type, part of ring, stereo configuration, aromaticity). Converting these features (except for atom mass) into one-hot encoded vectors resulted in a feature vector with $F_b = 10$ dimensions for every bond and in a feature vector with $F_a = 32$ dimensions for every atom.

For a substrate with $N$ atoms, we stored all bonds in an $N \times N$-dimensional adjacency matrix $A$, i.e., entry $A_{vw}$ is equal to 1 if there is a bond between the 2 atoms $v$ and $w$ and 0 otherwise. We stored the bond features in a $(N \times N \times F_b)$-dimensional tensor $E$, where entry $E_{vw} \in \mathbb{R}^{F_b}$ contains the feature vector of the bond between atom $v$ and atom $w$. Afterwards, we expanded tensor $E$ by concatenating the feature vector of atom $v$ to the feature vector $E_{vw}$. If there was no bond between the atoms $v$ and $w$, i.e., $A_{vw} = 0$, we set all entries of $E_{vw}$ to zero. We then used the resulting $(N \times N \times (F_a + F_b))$-dimensional tensor $E$, together with the adjacency matrix $A$, as the input of the GNN.

During training, the number of atoms $N$ in a graph has to be restricted to a maximum. We set the maximum to 70, which allowed us to include most of the substrates in the training. After training, the GNN can process substrates of arbitrary sizes.

### Architecture of the graph neural network

In addition to the predefined fingerprints, we also used a GNN to represent the substrate molecules. We first give a brief overview over such GNNs, before detailing our analysis.

As in the calculations of the ECFPs, a molecule is represented as a graph by interpreting the atoms as nodes and the chemical bonds as edges. Before a graph is processed by a GNN, feature vectors $\overrightarrow{x}_v$ for every node $v$ and feature vectors $\overrightarrow{e}_{vw}$ for every edge between 2 nodes $v$ and

*w* are calculated. We calculated 8 features for every atom and 4 features for every bond of a substrate, including mass, charge, and type of atom as well as type of bond (see Methods, "Calculation of the input of the graph neural network"). The initial representations $\vec{x}_v = \vec{x}_v^{(0)}$ and $\vec{e}_{vw} = \vec{e}_{vw}^{(0)}$ are updated iteratively for a predefined number of steps $T$ using the feature vectors of the neighboring nodes and edges (**Fig 1B**). During this process, the feature vectors are multiplied with matrices with trainable entries, which are fitted during the optimization of the GNN. After $k$ iterations, each node representation $\vec{x}_v^{(k)}$ contains information about its k-hop neighborhood graph. After completing $T$ iteration steps, all node representations are averaged to obtain a single vector $\vec{x}$, which represents the entire graph [43,44]. The vector $\vec{x}$ can then be used as an input of an FCNN to predict properties of the graph (the $K_M$ value of the molecule in our case; **Fig 1**).

The described processing of a graph with a GNN can be divided into 2 phases. The first, message passing phase consists of the iteration process. The second, readout phase comprises the averaging of the node representations and the prediction of the target graph property [43]. During the training, both phases are optimized simultaneously. The vector $\vec{x}$ can thus be viewed as a task-specific fingerprint of the substrate. Since the model is trained end to end, the GNN learns to store all information necessary to predict $K_M$ in this vector [44,45].

We use a variant of GNNs called directed message passing neural network (D-MPNN) [22,46]. In D-MPNNs, every edge is viewed as 2 directed edges pointing in opposite directions. During the iteration process (the message passing phase), feature vectors of nodes and edges are iteratively updated. To update them, feature vectors of neighboring nodes and edges are multiplied by matrices with learnable parameters and the results are summed. Then, an activation function, the *ReLU*, is applied to the resulting vector to introduce nonlinearities.

We set the number of iterations for updating the feature vector representations to $T = 2$. The dimension of the feature vectors during the message passing phase are set to $D = 50$. We apply batch normalization before every activation function. Additionally, we tried to apply dropout at the end of the message passing phase, but this does not improve model performance.

After the message passing phase, the readout phase starts, and feature vectors of all nodes and edges are pooled together using an order-invariant function to obtain a single vector $\vec{x} \in \mathbb{R}^D$, which is a representation of the input. The pooling is done using the element-wise mean of the feature vectors. We then concatenate $\vec{x}$ with the *MW* and the *LogP*, which are global molecular features that are correlated with the $K_M$ value [28]. This results in an extended fingerprint $\vec{\hat{x}} = (\vec{x}^\top, MW, LogP)^\top \in \mathbb{R}^{D+2}$.

Afterwards, $\vec{\hat{x}}$ is used as the input of an FCNN with 2 layers with dimensions 32 and 16, again using *ReLUs* as activation functions. Batch normalization and L2-regularization are applied to the fully connected layers to avoid overfitting.

During training, the values of the matrices from the message passing phase and the parameters of the FCNN from the readout phase are fitted simultaneously. We trained the model by minimizing the *MSE* with the optimizer Adadelta [47] with a decaying learning rate (decay rate to $\rho = 0.95$), starting at 0.05 for 50 epochs. We used a batch size of 64, a regularization parameter $\lambda = 0.01$ for the parameters in the message passing phase, and a regularization parameter $\lambda = 1$ for the parameters in the readout phase. The hyperparameters regularization factor, learning rate, batch size, dimension of feature vectors $D$, and decay rate were optimized with a 5-fold cross-validation on the training set by performing a grid search. We selected the set of hyperparameters with the lowest mean *MSE* during cross-validation.

## UniRep vectors

To obtain a 1,900-dimensional UniRep vector for every amino acid sequence in the dataset, we used Python code that is a simplified and modified version of the original code from the George Church group [24] and which contains the already trained UinRep model (available from https://github.com/EngqvistLab/UniRep50). The UniRep vectors were calculated from a file in FASTA format [48], which contained all amino acid sequences of our dataset.

## Fitting of the gradient boosting model with substrate and enzyme information

We concatenated the task-specific substrate fingerprint $\overrightarrow{x} \in \mathbb{R}^{52}$ and the 1,900-dimensional UniRep vector with information about the enzyme's amino acid sequence. We used the resulting 1,952-dimensional vector as the input for a gradient boosting model for regression, which we trained to predict the $K_M$ value. We set the maximal tree depth to 7, minimum child weight to 10.6, maximum delta step to 4.24, the learning rate to 0.012, the regularization coefficient $\lambda$ to 3.8, and the regularization coefficient $\alpha$ to 3.1. We trained the model for 1,381 iterations. The hyperparameters regularization coefficients, learning rate, maximal tree depth, maximum delta step, number of training iterations, and minimum child weight were optimized by performing a grid search during a 5-fold cross-validation on the training set. We selected the set of hyperparameters with the lowest mean *MSE* during cross-validation.

## Model comparison

To test if the differences in performance between the models with predefined fingerprints as input and the model with the task-specific fingerprint as input are statistically significant, we applied a one-sided Wilcoxon signed-rank test. The Wilcoxon signed-rank test tests the null hypothesis that the median of the absolute errors on the test set for predictions made with the model with task-specific fingerprints, $\bar{e}_1$, is greater or equal to the corresponding median for predictions made with a model with predefined fingerprints, $\bar{e}_2$ ($H_0 : \bar{e}_1 \geq \bar{e}_2$ versus $H_1 : \bar{e}_2 > \bar{e}_1$). We could reject $H_0$ ($p = 0.0022$ (ECFP), $p = 0.0515$ (RDKit), $p = 0.030$ (MACCS keys)), accepting the alternative hypothesis $H_1$.

Analogous to the described procedure, we tested if the difference in model performance between the GNNs with and without the 2 additional features, *MW* and *LogP*, is statistically significant. We could reject the null hypothesis $H_0$ that the median of the absolute errors on the test set for predictions made with the GNN with *MW* and *LogP* is greater or equal to the corresponding median for predictions made with the GNN without additional feature ($p = 0.0454$). To execute the tests, we used the Python library SciPy [49].

## Prediction of $K_M$ values for genome-scale models

We downloaded 46 genome-scale models from BiGG [50] and the genome-scale model yeast8 for *S. cerevisiae* [51]. We extracted all enzymatic reactions from these models and created 1 entry for every substrate in an enzymatic reaction. We extracted the KEGG Compound IDs for every substrate from the annotations of the model, if available; otherwise, we mapped the substrate names to KEGG Compound IDs via synonym lists from KEGG and PubChem in the same way as described for the substrate names in the BRENDA and Sabio-RK datasets. To obtain the enzyme information, we used the gene reaction rules, which contain the names of the involved genes. To obtain the amino acid sequence and the UniProt ID for every enzyme, we used the UniProt mapping service [38]. If multiple enzymes are given for one reaction, we made a prediction for all of the given enzymes. If an enzyme complex consisted of multiple

genes, we tried to figure out which of the genes has a binding activity. Therefore, we downloaded for all of the associated UniProt IDs the GO annotations via QuickGO [52]. For every UniProt ID, we checked if a binding activity was stated in the annotations. If we found a binding activity for more than 1 UniProt ID or for none of the UniProt IDs in the enzyme complex, we did not use any enzyme information.

If enzyme and substrate information was available, we used the full model to predict $K_M$. If only substrate or only enzyme information was available, we used a gradient boosting model that only uses substrate or enzyme information as its input. If neither substrate nor enzyme information were available, we used the geometric mean over all $K_M$ values in the BRENDA dataset as a prediction.

To train the gradient boosting model to predict $K_M$ values, we used the whole BRENDA dataset for model training, including the test set.

## Supporting information

**S1 Table. Results of the hyperparameter optimizations of fully connected neural networks (FCNNs), which were trained to predict $K_M$ from substrate information only.** The hypeparameter optimizations were performed for each of 4 different fingerprints of the substrates with a 5-fold cross-validation on the training set.
(TIF)

**S2 Table. Results of the hyperparameter optimizations of gradient boosting models, which were trained to predict $K_M$ from substrate information only.** The hypeparameter optimizations were performed for each of 4 different fingerprints of the substrates with a 5-fold cross-validation on the training set.
(TIF)

**S3 Table. Results of the hyperparameter optimizations of elastic nets, which were trained to predict $K_M$ from substrate information only.** The hyperparameter optimizations were performed for each of 4 different fingerprints of the substrates with a 5-fold cross-validation on the training set.
(TIF)

**S4 Table. Result of the hyperparameter optimization of a fully connected neural networks (FCNN), which was trained to predict $K_M$ from substrate and enzyme information (GNN fingerprint and UniRep vector).** The hypeparameter optimization was performed with a 5-fold cross-validation on the training set.
(TIF)

**S5 Table. Result of the hyperparameter optimization of the gradient boosting model, which was trained to predict $K_M$ from substrate and enzyme information (GNN fingerprint and UniRep vector).** The hypeparameter optimization was performed with a 5-fold cross-validation on the training set.
(TIF)

**S6 Table. Result of the hyperparameter optimization of an elastic net, which was trained to predict $K_M$ from substrate and enzyme information (GNN fingerprint and UniRep vector).** The hyperparameter optimization was performed with a 5-fold cross-validation on the training set.
(TIF)

**S1 Fig. Scatter plot of log10-transformed $K_M$ values predicted with the gradient boosting model with substrate and enzyme information as inputs versus the experimental values**

**downloaded from BRENDA and Sabio-RK.** The scatter plot displays all data points of the Sabio-RK test set (orange) and all data points from the BRENDA test set with an EC number–substrate combination not present in the training set (violet). The data underlying the graphs shown in this figure can be found at https://github.com/AlexanderKroll/KM_prediction/tree/master/figures_data.
(TIF)

**S2 Fig. (a)** Scatter plot of measured $K_M$ values and the absolute prediction errors of the BRENDA test data points for which neither the substrate nor the enzyme occurs in the training set. (b) Histogram with the distribution of the $K_M$ values in the training set. The data underlying the graphs shown in this figure can be found at https://github.com/AlexanderKroll/KM_prediction/tree/master/figures_data.
(TIF)

**S1 Data. Dataset in xlsx format containing complete $K_M$ predictions for 47 genome-scale metabolic models, including those for *Homo sapiens*, *Mus musculus*, *Saccharomyces cerevisiae*, and *Escherichia coli*.**
(XLSX)

## Acknowledgments

We thank Hugo Dourado, Markus Kollmann, and Kyra Mooren for helpful discussions. Computational support and infrastructure was provided by the "Centre for Information and Media Technology (ZIM) at the University of Düsseldorf (Germany).

## Author Contributions

**Conceptualization:** Alexander Kroll, David Heckmann, Martin J. Lercher.

**Data curation:** Alexander Kroll.

**Formal analysis:** Alexander Kroll.

**Funding acquisition:** Martin J. Lercher.

**Investigation:** Alexander Kroll.

**Methodology:** Alexander Kroll, David Heckmann.

**Software:** Alexander Kroll, Martin K. M. Engqvist.

**Supervision:** Martin J. Lercher.

**Validation:** Alexander Kroll.

**Visualization:** Alexander Kroll.

**Writing – original draft:** Alexander Kroll.

**Writing – review & editing:** Alexander Kroll, Martin K. M. Engqvist, David Heckmann, Martin J. Lercher.

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
