## [Editor Report · Decision Letter 0]

17 Dec 2020

Dear Dr Lercher, 

Thank you for submitting your manuscript entitled "Prediction of Michaelis constants from structural features using deep learning" for consideration as a Short Report by PLOS Biology.

Your manuscript has now been evaluated by the PLOS Biology editorial staff, as well as by an academic editor with relevant expertise, and I'm writing to let you know that we would like to send your submission out for external peer review.

IMPORTANT: We note that you submitted your paper as a Short Report, but we think it would be better considered as a Methods and Resources paper. No re-formatting is needed, but please could you change the article type to "Methods and Resources" when you upload your metadata (see next paragraph)?

Please re-submit your manuscript within two working days, i.e. by Dec 21 2020 11:59PM.

Kind regards,

Roli Roberts

Senior Editor

PLOS Biology

---

## [Decision Letter · Decision Letter 1]

10 Mar 2021

Dear Dr Lercher,

Thank you very much for submitting your manuscript "Prediction of Michaelis constants from structural features using deep learning" for consideration as a Research Article at PLOS Biology. Your manuscript has been evaluated by the PLOS Biology editors, an Academic Editor with relevant expertise, and by five independent reviewers. I must first apologise for the extraordinary time that it has taken us to return a decision to you, and for the unusual number of reviewers (we usually aim for three or four). These two issues are both related to difficulties that we encountered in recruiting reviewers during these difficult times.

However, we think that despite the five reviews, there is considerable overlap between reviewers #1 and #2 and between reviewers #3, #4 and #5, so we think that the revision task will not be proportionately arduous. After discussion with the Academic Editor, we provide the following guidance:

IMPORTANT: While we recognise the reasons behind the negative assessments of reviewers #1 and #2, we also see the potential that is perceived by reviewer #3, #4 and #5. On that basis, we have decided to invite a revision. However, you should ensure that you address some of the points raised by reviewers #1 and #2 by clarifying who/what the target market for this method/resource is, and why the large-scale but approximate prediction of Km is useful; you should also ensure that your coverage of the prior literature is balanced.

In terms of reviewers #3, #4 and #5, we see significant overlap, with the requests largely falling into the following categories:

a) perform more extensive cross-validation and sensitivity testing (#4, #5).

b) perform additional analyses (domain order for #3, use multiple empirical values rather than means for #4, different database and "fingerprint" method for #5).

c) provide a handy user interface (updatable website for #3 or Shiny app for #5).

d) make the article more accessible to a broader readership (#5).

We feel that categories "a" and "d" are essential, "c" is advisable (to maximise utility to the community, and uptake), and you should either attempt or reasonably rebut the analyses requested under "b" (most of which sound sensible to us).

In light of the reviews (below), we will not be able to accept the current version of the manuscript, but we would welcome re-submission of a much-revised version that takes into account the reviewers' comments. We cannot make any decision about publication until we have seen the revised manuscript and your response to the reviewers' comments. Your revised manuscript is also likely to be sent for further evaluation by the reviewers.

We expect to receive your revised manuscript within 3 months. 

**IMPORTANT - SUBMITTING YOUR REVISION**

*Re-submission Checklist*

*Published Peer Review*

*PLOS Data Policy*

*Blot and Gel Data Policy*

Sincerely,

Roli Roberts

Senior Editor,

rroberts@plos.org,

PLOS Biology

REVIEWERS' COMMENTS:

Reviewer #1:

This paper describes several machine learning approaches to predict Km values. The most successful of these has an R^2 value of 0.42 between predicted and measured values for a test dataset. The predictions do not take into account experimental conditions (pH, temperature, ionic strength). They also assume that each enzyme has just one Km but allosteric enzymes, which are very common, have multiple Km values (1 per site) although a midpoint concentration corresponding to 0.5 Vmax can still be defined. Consequently, I'm skeptical whether the method is of value for researchers studying a specific system (e.g. as in drug development) but it could be helpful for large-scale modelling of metabolic fluxes, for example, when the details might matter less. 

Reviewer #2:

In the manuscript "Prediction of Michaelis constants from structural features using deep learning" by Kroll et. al. a graph-based method of predicting Km for a ligand is illustrated utilizing both chemical descriptors as well as a binary vector of a protein domains. Broadly, the work done is executed well but conclusions are significantly overstated. As the authors state line ~70 "a machine learning model trained on domain structures will not be able to distinguish [specific molecule binding affinity]; the model will assign the same predictions for both substrates to each of the two proteins". The authors then go and make a model based on domains. While I agree this model will provide a likely generic estimate for a likely biological revenant ligand based on the chemical structure (which is understandable given the dominance of chemical structure in feature importance) it is not going to be predictive of a molecule's affinity to the protein as it will not account for the effects mutations have within a binding site or tertiary effects. It simply will predict the average likely Km level for the class of proteins, which generally can be extracted simply from a simple look-up table where the average Km for a domain annotated (or slightly more elaborated version where MW or LogP or such are incorporated). Furthermore, the authors seem to have either ignored or been unaware of the decades of research in the field of affinity prediction, the following review (I am not an author or affiliated with any of the authors) nicely summarizes:

Comparison Study of Computational Prediction Tools for Drug-Target Binding Affinities

https://www.frontiersin.org/articles/10.3389/fchem.2019.00782/full

A simple google search "predicting ligand affinity" or "predicting small molecule binding" or related identifies hundreds of manuscripts using ML to predict Ki/Kd/IC50 which in all essence are equivalent to Km being the pseudo-binding constant of an enzyme.

Therefore, while this article is interesting and contributes to the field, it should do a better job placing the advance within the field, make more accurate claims about the utility and advance, and is likely more appropriate for a trade journal. I do apologize if this review seems overly harsh, however I was truly excited when I read the title and abstract and found myself deeply frustrated by the time I had concluded reading the article.

Reviewer #3:

In this manuscript, the authors aimed to predict the values of Km from structures of the substrate and the enzyme. They represented the substrate structure with a graph neural network, accounting for comprehensive information of the atoms and their interactions with neighbors. For the enzyme structure, however, they used a much simpler representation: what functional domains exist in the enzyme. Using training data set from the BRENDA database, the authors were able to make predictions of Km values. The results were better than a previous work (Borger et al. 2006). 

Given the huge variation of Km values in the BRENDA database - for a typical enzyme in BRENDA, the Km value deviates 3.7-fold from the mean of all the values for that enzyme - the predicted values here on average deviate 4.3-fold and thus seem valuable. Furthermore, with the readily obtainable results from the computational model and the expandability of the model, I think this manuscript is an important contribution to understanding enzymatics at the systems level. I however do have some issues that I like the authors to address.

1. The authors present the model as the main result. However, to most biologists including me, the predicted values of Km are probably most valuable. I thus suggest the authors apply the model to all possible reactions in popular organisms such as human and mouse. Ideally, the results can be made available in the format of website (which can be updated as the model is updated in future). At least an excel file containing the numbers should be included in the supplementary materials. 

2. The representation of the enzyme structure seems over-simplified. The author discussed a number of enzyme properties that can be considered when they become more widely available in future. The order of the domains (along the protein sequence) however is available. I wonder if this structurally important information can be incorporated into the model? 

3. Can the authors discuss the applicability of the model to the prediction of Kcat? 

4. The origins of the Y-axis of the MSE and R2 figures were randomly chosen at the moment. I suggest set all of them to 0. 

Reviewer #4:

[please see attachment for correctly formatted version]

The manuscript by Kroll et al. uses a combination of machine and deep learning to predict Km values based on available information on substrate fingerprints. Currently, machine and deep learning are very “hot” topics in all fields. Their contribution to parameter prediction is of particular value to the systems biology community as it offers significant assistance in model parametrization.

The framework seems to be well researched and structured and the presentation of the paper is very good. The results also look very promising for Km prediction and it would indeed be very interesting to see results for Kcat parameters in the future. Also the github page with the model information is very practical and very well detailed.

A few points should be addressed before acceptance.

Major issues

1) The test procedure seems a little basic as there is only one test dataset. A full kfold cross validation would be better. 

2) It would also be nice to get some empirical investigation as to how robust the results are to different choices of train/validation/test data. It would be interesting to see how much the hyperparameters change, whether the errors are stable etc. 

Both of these issues could be addressed fairly simply by re-running the analysis on different splits. 

3) The choice about processing the input data (averaging log kms for enzyme/substrate combinations with the same functional domain) isn't well argued for and may have biased the results somewhat. A mentioned in lines 83-86: “Since we are only interested in one KM value for every substrate in combination with a group of enzymes with the same functional domain content (see below), we took the geometric mean if multiple values existed for the same combination of substrate and enzyme domain content.”

Not sure if this is a good idea - why throw away information like this? The mean is only a good summary statistic if the error distribution is approximately normal (which probably isn't the case here given that BRENDA data tends to be quirky) and even in this case the fact that certain data points represent more observations should be taken into account. Based on the arguments in the paper I think it would have been better to omit this step and allow multiple measurements of the same enzyme/substrate combination.

4) *** line 375: “We tested the null hypothesis that the median of the absolute errors on the test set...” 

Why median absolute error not MSE?

Minor issues

It would be better to package the dataset that was used for the analysis in the paper instead of downloading it from BRENDA for reproducibility purposes. As BRENDA changes periodically, some of the information may be different and it might not be possible to obtain the same results.

Reviewer #5:

In this manuscript, Kroll et al. describe and train a model to predict Km using machine learning. The authors are correct in presenting and highlighting the problem around the availability and the challenges of calculating km comprehensively, consistency and at scale. I also agree that the correct prediction of Km at scale will enormously benefit other fields so the impact could be huge. I however have significant methodological questions and I am not fully convinced that PLoS Biology is the ideal home for this manuscript. I think the manuscript on its present form is very technical and I am not sure of interest to the broad readership of PLoS Biology. My advice would be to perform major changes to adapt it to a wider audience or to submit to a more technical journal such as PLoS Computational Biology or j chem inf mod. The article is well written and relatively easy to follow. Below I detail my comments and questions:

Regarding the training/test data, the authors use BRENDA which quite a standard database. I however don't see the data being available as supplementary. I understand that there is a script in github to do the preprocessing but since BRENDA can be updated I would make the data preprocessed available as a supplementary for better reproducibility and future comparisons. This would also be of help to the non-computational readers of PLoS Biology. I was a bit surprised to see that the authors don't use also SABIO-RK database, that seems quite a standard in the field. On a quick search they don't seem to include exactly the same information so I wonder why the authors did not use it for model building or testing (see comment below). Finally, I was a bit surprised why the authors don't doo 5- or 10-fold cross validation although I have more questions regarding model testing below.

I think it was a wise decision to use parameters that are broadly applicable to aim for a method that is applicable to any enzyme of any species and a strength as compared to other models that use information that is not broadly available. I liked the idea of using ECFPs and RDKit is indeed a very good standard in chemoinformatics. However, there are many different types of fingerprints so it seems a wasted opportunity to not look a little at other fingerprint approaches. I would use at least another fingerprint to compare if you see any differences. In my opinion 2D-pharmacophore fingerprints have shown quite robust results and I think it would be interesting to see if better results are achieved... But if the authors would prefer another fingerprint to compare I would have no objections, I just think it would be nice to see at least another fingerprint tested.

Regarding model building, the authors jump straight away to using FCNNs. I think it would be very interesting to see how NNs compare to simpler models. So I would start by implementing a simpler model before jumping to neural networks. Maybe linear modelling (e.g. elastic search) would be a good place to start to be able to assess the added value of using more complex models.

The description of the results on graph neural networks was a bit technical. I wonder if it would be too much for a wet-lab reader of PLoS Biology... I would consider simplifying it and/or moving part of it to the results and in the main text focus more on the outcomes of the model and its applicability.

I liked how information was subsequently added on molecular weight, partition coefficient and functional domain to assess the effects in the model.

The authors then describe how to predict km for substrates and enzymes not present in the training data. And I get slightly confused and this is highlights the importance of providing the data that was used for model building and testing as a supplementary (see above). The numbers discussed seem really low. Only 3 homologous enzymes with the same substrate were present in the training and test set? And only 9 where the enzyme and substrate are not in the training? then additional trainings were created but again 45 points seems low to me, given there are 5,000 data points. So I think this needs careful design from the outset and more clear explanation. I think the authors should do x-fold cross validation of all the models and these test/training sets should be designed to prevent the same enzymes/homologues and substrates to be present in both training and test sets. That would be the fair approach to testing it in my opinion.

I appreciate the authors discuss the MSE differences with previous approaches despite not being possible to do a comprehensive comparison. I wonder why they don't discuss R2 too and there are a couple of previous works that are also not included in this discussion. I think this part should be elaborated more. 

In an ideal world, I would like to see a prospective experimental validation at the end of the paper. I think this is important, particularly if the aim is a general biology journal as PLoS Biology as opposed to a computational journal. I appreciated this may be difficult as measuring Km is not simple and the authors are fully computational. But I wonder if the authors could alleviate this somehow. Maybe using the SABIO-RK data that is not covered by BRENDA as an unseen dataset? Or maybe curation some recent Km literature values that did not make it into BRENDA yet? I think this would be important.

Finally, I think there are two aspects that would make this far more appealing for a broad biology journal. These would not be necessary for a computational journal but I think they are for PLoS Biology. The first one is data availability. I know the code is in github but experimental scientists that could benefit from your model can't access this easily. I think you should provide the Km predicted with your best model for all enzymes. Of course this not easy to do but maybe you could calculate them for all the enzymes in the most used models and release these data. Alternatively, my preferred option would be that you somehow enabled wet-lab scientist to be able to calculate the km themselves using your method with an easy online application, for example deploying a shiny app. The second aspect is that the paper is very dry on its current form. Could you maybe end with a simple example of use? Maybe an example where the Km is not known for am enzyme and this makes it very challenging to generate a simple model of cellular metabolism. But when you estimate the km using your best model, then you can easily construct a simple metabolism model for this enzyme that explain experimental observations. This is just a suggestion but ending the paper with an example of use would make if far less dry and more palatable for non-expert experimental scientists.

In summary, have enjoyed reading the manuscript and the article is clearly a step forward in Km prediction. I therefore strongly encourage the authors to perform the major revisions suggested or transfer the paper to a computational journal with fewer revisions needed.

---

## [Decision Letter · Decision Letter 2]

9 Aug 2021

Dear Dr Lercher,

Thank you for submitting your revised Methods and Resources entitled "Prediction of Michaelis constants from structural features using deep learning" for publication in PLOS Biology. I've now obtained advice from three of the original reviewers and have discussed their comments with the Academic Editor. 

Based on the reviews, we will probably accept this manuscript for publication, provided you satisfactorily address the following data and other policy-related requests.

IMPORTANT:

a) Please make your title more appealing by including an active verb and alluding to the scale of prediction that your approach allows. We suggest something like "Deep learning allows genome-scale prediction of Michaelis constants from structural features"

b) Please address my Data Policy requests below; specifically, please supply numerical values underlying Figs 2AB, 3AB, 4ABC, 5AB, S1, and cite the location of the data clearly in each relevant Fig legend.

We expect to receive your revised manuscript within two weeks. 

*Published Peer Review History*

*Early Version*

Sincerely,

Roli Roberts

Senior Editor,

rroberts@plos.org,

PLOS Biology

DATA POLICY:

Regardless of the method selected, please ensure that you provide the individual numerical values that underlie the summary data displayed in the following figure panels as they are essential for readers to assess your analysis and to reproduce it: Figs 2AB, 3AB, 4ABC, 5AB, S1. NOTE: the numerical data provided should include all replicates AND the way in which the plotted mean and errors were derived (it should not present only the mean/average values).

DATA NOT SHOWN?

REVIEWERS' COMMENTS:

Reviewer #3:

The authors have satisfactorily addressed all my concerns. 

Reviewer #4:

All points were answered successfully and I have no further comments. 

Reviewer #5:

I thank the authors for their hard work addressing nearly all the comments from the 5! reviewers. I think they have done a good job, the paper is now much stronger and I have enjoyed (and learned!) reading the revised version of the manuscript. I therefore recommend this paper for publication. I would finally encourage the authors to apply the model that they have developed, maybe by starting collaborations with metabolite flux scientists that are dry or wet lab. We need good models need to get out there and be used! Congratulations.

---

## [Editor Report · Decision Letter 3]

26 Aug 2021

Dear Dr Lercher,

On behalf of my colleagues and the Academic Editor, Jason Locasale, I'm pleased to say that we can in principle offer to publish your Methods and Resources paper "Deep learning allows genome-scale prediction of Michaelis constants from structural features" in PLOS Biology, provided you address any remaining formatting and reporting issues. These will be detailed in an email that will follow this letter and that you will usually receive within 2-3 business days, during which time no action is required from you. Please note that we will not be able to formally accept your manuscript and schedule it for publication until you have made the required changes.

PRESS: We frequently collaborate with press offices. If your institution or institutions have a press office, please notify them about your upcoming paper at this point, to enable them to help maximise its impact. If the press office is planning to promote your findings, we would be grateful if they could coordinate with biologypress@plos.org. If you have not yet opted out of the early version process, we ask that you notify us immediately of any press plans so that we may do so on your behalf.

Sincerely, 

Roli Roberts

Roland G Roberts, PhD 

Senior Editor 

PLOS Biology

rroberts@plos.org